# Insights into the mechanism of coreactant electrochemiluminescence facilitating enhanced bioanalytical performance

Alessandra Zanut[1,5], Andrea Fiorani [1,6], Sofia Canola [1], Toshiro Saito[2], Nicole Ziebart[3], Stefania Rapino [1], Sara Rebeccani [1], Antonio Barbon [4], Takashi Irie[2], Hans-Peter Josel[3], Fabrizia Negri [1], Massimo Marcaccio [1], Michaela Windfuhr[3], Kyoko Imai[2], Giovanni Valenti [1✉] & Francesco Paolucci [1✉]

Electrochemiluminescence (ECL) is a powerful transduction technique with a leading role in the biosensing field due to its high sensitivity and low background signal. Although the intrinsic analytical strength of ECL depends critically on the overall efficiency of the mechanisms of its generation, studies aimed at enhancing the ECL signal have mostly focused on the investigation of materials, either luminophores or coreactants, while fundamental mechanistic studies are relatively scarce. Here, we discover an unexpected but highly efficient mechanistic path for ECL generation close to the electrode surface (signal enhancement, 128%) using an innovative combination of ECL imaging techniques and electrochemical mapping of radical generation. Our findings, which are also supported by quantum chemical calculations and spin trapping methods, led to the identification of a family of alternative branched amine coreactants, which raises the analytical strength of ECL well beyond that of present state-of-the-art immunoassays, thus creating potential ECL applications in ultra-sensitive bioanalysis.

[1] Department of Chemistry Giacomo Ciamician, University of Bologna, via Selmi 2, 40126 Bologna, Italy. [2] Hitachi High-Tech Corporation, 882, Ichige, Hitachinaka-shi, Ibaraki-ken 312-8504, Japan. [3] Roche Diagnostics GmbH, Nonnenwald 2, 82377 Penzberg, Germany. [4] Department of Chemical Sciences, University of Padova, Via F. Marzolo 1, 35131 Padova, Italy. [5] Present address: Tandon School of Engineering, New York University, Brooklyn, NY 11201, USA. [6] Present address: Department of Chemistry, Keio University, 3—14—1 Hiyoshi, Yokohama 223—8522, Japan. ✉email: g.valenti@unibo.it; francesco. paolucci@unibo.it

Diagnostic markers, or biomarkers, are biomolecules (e.g., enzymes, proteins, peptides, and hormones) that can be measured accurately and reproducibly and can precisely predict relevant clinical outcomes or diseases in various populations[1,2]. In fact, biomarkers represent a powerful aid in clinical diagnostic and therapeutic monitoring. Therefore, detection, identification, and quantification of such molecules can translate into the development of sophisticated methods and instrumentations for analyzing clinically useful biomarkers[3–6]. In particular, research into noninvasive and sensitive methods and strategies plays a pivotal role in early diseases diagnosis and treatment[7,8]. One of the main challenges, however, is the ultralow concentrations (i.e., picomolar or below) of biomarkers in the human body and in complex matrices, such as blood, urine, and tissues[9–11]. In this context, electrochemiluminescence (ECL) appears to be a leading transduction technique thanks to the optimal combination of electrochemical and spectroscopic methods. ECL has received enormous attention as a powerful tool in the biosensing field. Despite its high sensitivity, however, ECL remains intrinsically a surface-confined process that incorporates concomitant steps to eventually generate the analytical signal[12–16]. Optimization of such a mechanism is still underway and is of fundamental importance to several highly promising applications aimed at the quantification of crucial biomarkers. In all cases, the rate-determining steps for the overall process include (i) the kinetics underlying the heterogeneous electron transfer reactions of the coreactant, (ii) the stability of coreactant radicals, and (iii) the distribution of ECL luminophores. These factors critically affect the ECL mechanism and the final signal efficiency[16,17]. Here, we identify a mechanism for boosting ECL emission by optimizing the distribution of radicals and luminophores; moreover, we demonstrate the possibility of enhancing signals for quantifying important markers. In fact, ECL possesses unique advantages, such as (i) a superior temporal and spatial control of light emission, (ii) very low background signal and high sensitivity because of the absence of excitation light, and (iii) a broad dynamic range and rapid measurements of small volumes. Altogether, these characteristics make ECL, and in particular the coreactant approach[18], the most used transduction methodology, particularly for applications in clinical assays using highly complex matrices, such as urine, blood, or lysates[19–22].

The first description by Bard and coworkers regarding the coreactant mechanism in heterogeneous ECL (i.e., luminophores are not free to diffuse, Fig. 1,[23]) was a fundamental breakthrough in the development of analytical ECL applications[23]. In their pioneering research, the authors investigated an alternative ECL pathway using tri-*n*-propylamine (TPrA) as the sacrificial oxidative-reduction coreactant and tris(2,2′-bipyridine)ruthenium(II) ([Ru(bpy)$_3$]$^{2+}$) as the emitting species[23,24]. Their findings, combined with simulations reported by Wightman and coworkers[25], opened a wide window in the literature concerning the applications of ECL in the field of biosensors. As a result, an exponential increase in the number of publications in this area of knowledge has been observed in recent years[26]. This transformed ECL from an academic curiosity to a real application and industrial success. However, to date, the mechanism of ECL generation proposed by Bard and coworkers remains the only one accepted, and TPrA is the most efficient ECL coreactant for applied research[27].

In commercial ECL-based immunoassays, such as Elecsys® immunoassays[28], biomarkers are detected after their immobilization on the working electrode via magnetic beads, which are attracted to the electrode surface through a magnet. Typically, microbeads with a diameter of 2.8 μm are used in these assays. ECL emission is not homogeneous on the bead surface because of the spatial distribution of the electrogenerated TPrA radicals

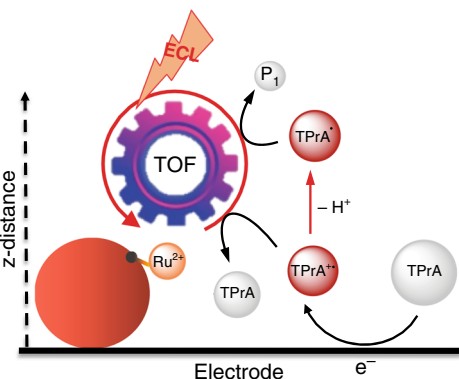

**Fig. 1 Schematic representation of the remote electrochemiluminescence (ECL) mechanism.** Tri-*n*-propylamine (TPrA) is oxidized at the electrode, generating the radical cation (TPrA$^{•+}$), which rapidly [half-life ($t_{1/2}$), ~200 μs] deprotonates, forming the radical (TPrA$^{•}$). The radical and radical cation react with the ECL luminophore [Ru(bpy)$_3$]$^{2+}$, tris(2,2′-bipyridine)ruthenium(II) (Ru$^{2+}$), on the magnetic beads (red sphere), herein named surface generation–bead emission. The turnover frequency (TOF) is the number of photons emitted in 1 s by a single luminophore (see "Supplementary Methods 1").

(Fig. 1). In particular, the TPrA radical cation, with its limited lifetime [half-life ($t_{1/2}$), ~200 μs], is not expected to diffuse farther than 3 μm from the electrode surface[29–32].

In this work, through the combination of ECL and microscopy[26,33,34] and the use of labeled microbeads, we map ECL generation close to the electrode surface (≤1 μm), thus revealing the contribution of an additional pathway to ECL generation, which was unobserved to date. This additional mechanism exhibits a very high efficiency, i.e., 10 times more intense than the signals measured at larger distances (>1 μm). Furthermore, inspired by the mechanistic hypothesis proposed to explain these findings, we identify a family of alternative coreactants/additives, namely branched amines, which may lead to an advantageous overall signal enhancement. In particular, the use of *N*-dipropyl isobutyl amine (DPIBA) enhances the ECL signal by a maximum of 47% in a commercial immunoassay system for the quantification of several biomarkers, such as thyroid stimulating hormone (TSH), cardiac troponin T, ferritin, and immunoglobulin (Ig)M antibodies against *Toxoplasma gondii* (Toxo IgM) and hepatitis A (A-HAV IgM).

## Results

**Surface generation–bead emission.** Based on the surface generation–bead emission configuration, we evaluated the effect of luminophore distribution on ECL generation using a series of beads with different sizes (Fig. 2a and "Supplementary Methods 1"). In this case, we collected ECL emitted from a single bead labeled with the [Ru(bpy)$_3$]$^{2+}$ luminophore (Ru@beads; "Supplementary Methods 1" and Supplementary Fig. 1) and positioned under a direct microscope equipped with an electron-multiplying charge-coupled device (EMCCD) camera (Fig. 2a). Streptavidin-coated polystyrene magnetic microbeads with different sizes (diameters of 2.8, 1, 0.5, and 0.3 μm) were covalently coupled to a ruthenium-containing antibody functionalized with biotin[35]. Comparisons of ECL signals of Ru@beads with different sizes allowed us to correlate ECL emission as a function of distance to the electrode.

**Turnover frequency.** ECL efficiency was quantified using combinations of different analytical techniques. ECL intensities were measured via imaging experiments (Fig. 2a) in which a potential

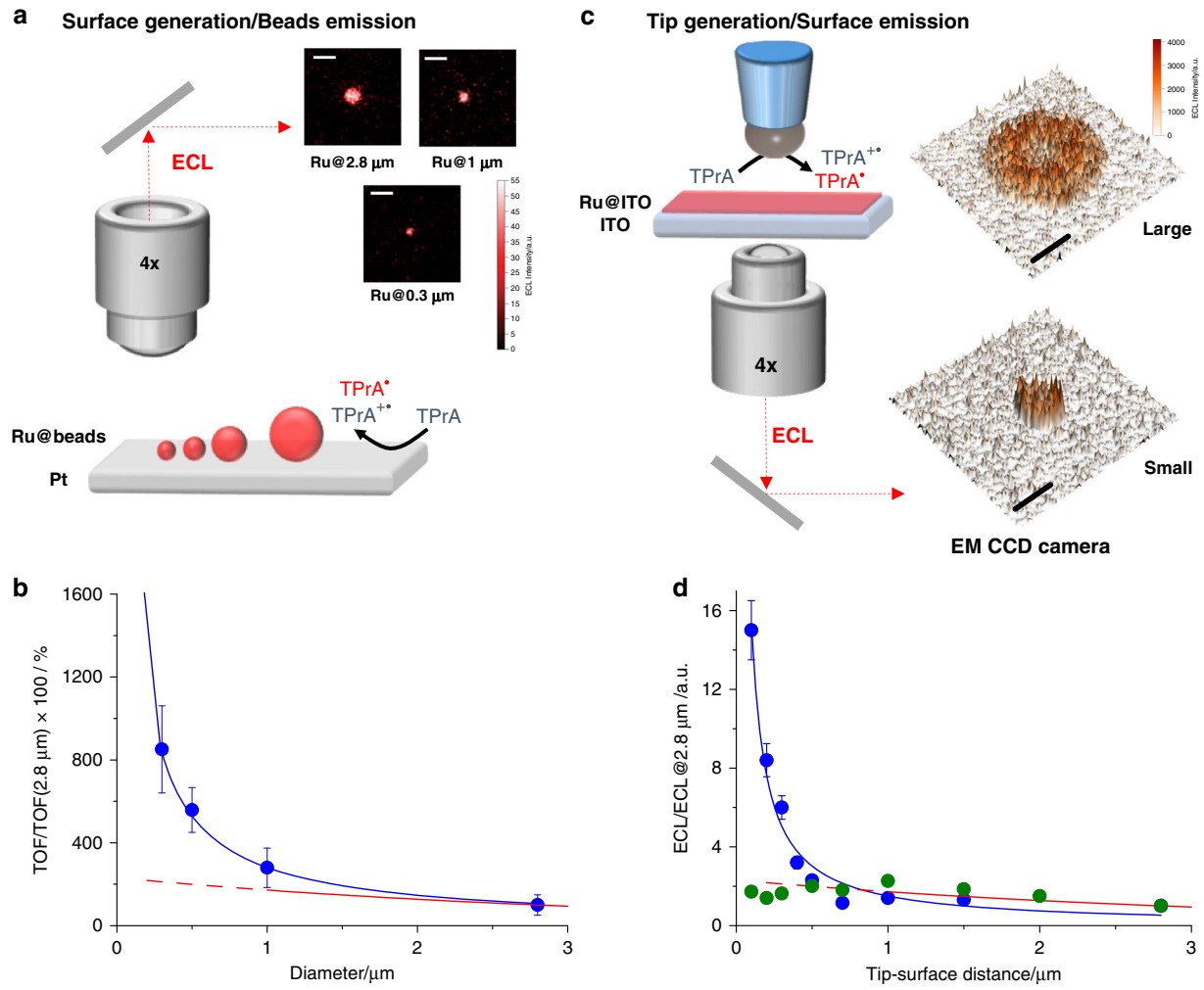

**Fig. 2 Spatial map of electrochemiluminescence (ECL) emission. a** Schematic representation of the surface generation–bead emission experiment where Ru@beads are magnetic beads labeled with $[Ru(bpy)_3]^{2+}$ on platinum electrode (Pt) and **b** turnover frequency (TOF) as a function of bead size (blue curve and dots) and ECL intensity of the ref. [23] (red curve). The inset in **a** shows the ECL images acquired for beads with different sizes where Ru@2.8 µm, Ru@ 1 µm and Ru@0.3 µm are ECL images of magnetic beads labeled with $[Ru(bpy)_3]^{2+}$ with a diameter of 2.8, 1, and 0.3 µm, respectively. Magnification ×100, Scale bar 5 µm; potential applied, 1.4 V (vs. Ag/AgCl, 3 M KCl); acquisition time, 0.5 s. The TOF is the number of photons emitted in 1 s by a single Ru luminophore (TOF = $(ECL_{Ru@Bead} - ECL_{Bead})/n°$ of $[Ru(bpy)_3]^{2+}$ × time see also Results and "Supplementary Methods 1"). TOF@2.8 µm is the turnover frequency for 2.8 µm beads (Ru@2.8 µm). **c** Schematic representation of the tip generation–surface emission experiment where Ru@ITO is transparent indium tin oxide (ITO) electrode functionalized with a $[Ru(bpy)_3]^{2+}$ monolayer as the emitting surface and **d** ECL intensity as a function of the tip–surface distance for a small electrode (blue curve and dots) and a large electrode (green dots) and ECL intensity of the ref. [23] (red curve). ECL@2.8 µm is the ECL intensity for 2.8 µm distance between tip and surface. The inset in **c** shows the ECL intensity image profile acquired for a small and a large Pt hemispherical electrode at 1.4 V. Magnification ×4, scale bar, 500 µm. The potentials applied via cyclic voltammetry at a scan rate of 100 mV s$^{-1}$ ranged from 0 to 1.4 V (vs. Ag/AgCl 3 M KCl). TPrA: tri-*n*-propylamine. Error bar shows the standard deviation ($n = 10$).

of 1.4 V (or 1 V, as in Supplementary Fig. 2) was applied for 0.5 s and integrated signals were obtained from the images. Turnover frequency (TOF)[36], as a function of bead size, was defined as the number of photons generated by 1 mol of luminophore per time unit and calculated using Eq. (1):

$$TOF = \frac{(ECL_{Ru@Bead} - ECL_{Bead})}{n° \ of \ [Ru(bpy)_3]^{2+} \times time} \quad (1)$$

where $ECL_{Ru@Bead}$ is the integrated ECL signal of a single bead, $ECL_{Bead}$ is the background (measured in the absence of [Ru (bpy)$_3$]$^{2+}$ luminophores), and $[Ru(bpy)_3]^{2+}$ is the amount of Ru luminophores. Ru luminophores were quantified via inductively coupled plasma mass spectrometry (ICP-MS) after sample mineralization (Supplementary Table 1 and "Supplementary Methods 1").

TOF, a new concept in ECL intensity quantification, allows the direct comparison of objects (i.e., beads) with different sizes. The ECL performance of small beads resulted in an outstanding TOF compared with that of large beads (Fig. 2b and Supplementary Figs. 3 and 4), thus confirming (i) the exponential increase in ECL emission close to the electrode surface and (ii) the far superior efficiency of 0.3-µm beads compared with that of 2.8-µm beads (8 times higher signal).

**Tip generation–surface emission.** In 2002, Bard and coworkers pioneered the investigation of ECL distribution in the proximity of the electrode surface using a hemispherical microelectrode controlled by a micromanipulator[23]. The principle of this experiment is that since ECL can only occur when both TPrA radical and TPrA radical cation are present and since TPrA radical cations show limited stability, the ECL intensity can be

effectively tuned by changing the tip–surface distance (Supplementary Fig. 3). Inspired by this approach, to investigate further the ECL efficiency at very short distances, we used a system comprising a transparent indium tin oxide (ITO) electrode functionalized with a $[Ru(bpy)_3]^{2+}$ monolayer as the emitting surface (Ru@ITO) and two different hemispherical Pt microelectrodes with a diameter of either 1.5 or 0.5 mm for TPrA oxidization positioned on an inverted microscope equipped with an EMCCD camera (Fig. 2c). This system was coupled with a micropositioner and was used to map ECL emission at different tip–surface distances while precisely controlling the distance between Ru@ITO and the microelectrode (Supplementary Figs. 5–8 and "Supplementary Methods 2")[37]. Upon application of proper potentials, microscopic inspection of the electrode surface evidenced the presence of an emitting disc associated with ECL generation underneath the microelectrode tip. In analysis of ECL intensity vs. tip–surface distances, relatively large distances (≥1 μm) resulted in an emission–distance profile consistent with that reported previously (Fig. 2d)[23].

However, as the 1.5-mm microelectrode tip was brought very close to Ru@ITO, the ECL emission intensity became unevenly distributed over the Ru layer facing the microelectrode tip. This resulted in a significantly lower emission from the inner part of the emitting disc and thus a lower overall ECL intensity (inset Fig. 2c, Supplementary Fig. 6, and Supplementary Movie 1). Such a behavior was not totally unexpected and was associated with the strong hindrance of TPrA diffusion to the central part of Ru@ITO under the microelectrode tip. Of note, and contrary to the effect on ECL intensity, the TPrA oxidation current was not affected by the microelectrode–surface distance because of the very large electrode used, which minimized the hindering effects on current generation (Supplementary Fig. 6). In addition, control experiments showed a non-negligible and constant contribution of the Pt electrode reflection to the light collected at different tip–surface distances (Supplementary Fig. 7). In line with the hypothesis presented above, the use of a smaller microelectrode (0.5 mm), which diminished the hindrance effects, resulted in a significant relative increase in ECL intensity at very short tip–surface distances (i.e., <1 μm) (Supplementary Fig. 8 and Supplementary Movie 2)[38].

Plotting the ECL intensity as a function of the tip–surface distance clearly indicated a trend similar to that reported previously for distances of >1 μm[23] but contrary to that reported for shorter distances. These different trends highlight the presence of a very efficient ECL generation mechanism operating very close to the electrode surface, which is in line with the experimental findings described in the previous sections (Fig. 2d).

Using the ECL intensity–distance dataset, we estimated the lifetime of radical species. For this, each tip–surface distance was converted to the corresponding (travel) time using Eq. (2), which is valid for planar diffusion (Supplementary Fig. 9).

$$t = \frac{d^2}{36D} \qquad (2)$$

where $d$ is the distance (in cm) between the tip and the Ru@ITO surface, and $D$ is the diffusion coefficient for TPrA ($7.4 \times 10^{-6}$ cm² s⁻¹)[23].

By analyzing the traveling time as a function of ECL intensity, we identified two different lifetimes: (i) at $d > 1$ μm, the analysis confirmed a half-life ($t_{1/2}$) of ~700 μs, which was attributed to the TPrA radical cation, as reported previously[23,25], and (ii) at smaller distances ($d < 1$ μm), where we observed a faster decay transient with a much shorter half-life ($t_{1/2}$, ~ 5 μs).

## Discussion

The prevailing ECL mechanism in biosensing and commercial immunoassay applications, called heterogeneous ECL, exclusively involves the radicals obtained by anodic oxidation of TPrA, during which all components of the immune complex, recognition unit, biomarker, and detection unit, labeled with the ECL luminophores, are situated close to the working electrode. Moreover, this mechanism involves direct oxidation of TPrA, which partially undergoes deprotonation (Fig. 1), thus forming a stable energetic radical species that reduces the ECL luminophore $[Ru(bpy)_3]^{2+}$ to $[Ru(bpy)_3]^+$. Conversely, the oxidized coreactant is continuously produced at the electrode surface and can thus react with $[Ru(bpy)_3]^+$ to generate the excited state $[Ru(bpy)_3]^{2+*}$. Finally, $[Ru(bpy)_3]^{2+*}$ relaxes to the ground state, generating the ECL emission.

$$TPrAH^+ \rightleftharpoons TPrA + H^+ \qquad (3)$$

$$TPrA - e^- \rightleftharpoons TPrA^{\bullet+} \qquad (4)$$

$$TPrA^{\bullet+} \rightleftharpoons TPrA^{\bullet} + H^+ \qquad (5)$$

$$TPrA^{\bullet} + [Ru(bpy)_3]^{2+} \rightleftharpoons P1 + [Ru(bpy)_3]^+ \qquad (6)$$

$$TPrA^{\bullet+} + [Ru(bpy)_3]^+ \rightleftharpoons TPrA + [Ru(bpy)_3]^{2+*} \qquad (7)$$

$$[Ru(bpy)_3]^{2+*} \rightarrow [Ru(bpy)_3]^{2+} + h\nu \qquad (8)$$

where $P1$ is the product of the homogeneous $TPrA^{\bullet}$ oxidation.

In this case, both the radical and radical cation must be present to generate the signal; therefore, it was expected that the use of smaller beads (diameter, <1 μm) would enhance ECL efficiency, as shown in the mechanism depicted in Fig. 1. In line with this concept, we used the surface generation–bead emission configuration to detect the exponential increase in ECL efficiency in terms of TOF with decreasing bead size. Unexpectedly, the increment shown was largely different from the predicted trend described previously by Bard and coworkers, with an 8-fold increase in ECL signal observed using 0.3-μm beads compared with that observed using 2.8-μm beads (see the comparison of the red signal and blue curve in Fig. 2b). In this mechanism, the pathway that involves direct oxidation of $[Ru(bpy)_3]^{2+}$ was negligible. In fact, the control experiments performed at a working potential lower than the luminophore oxidation potential (i.e., <1.2 V; Supplementary Fig. 2) resulted in almost identical TOF values (vs. bead size) compared with the experiments performed at 1.4 V. These observations are also in line with the results of our previous study, in which we estimated the effect of direct $[Ru(bpy)_3]^{2+}$ oxidation on ECL intensity at a very small distance (<9 nm) from the electrode surface alone[39].

In an attempt to explain this trend, we mapped the stability of TPrA radical cation using the so-called tip generation–surface emission approach. The half-life of TPrA radical cation was 700 μs, which is consistent with the previously reported values[23,29]. However, on analyzing ECL at short distances (<1 μm), we observed an additional transient with a shorter lifetime that involved other reactive species. In this case, the measured half-life of the electrogenerated species was 5 μs, suggesting a parallel mechanism triggered by TPrA oxidation. Analysis of products generated following prolonged electrolysis and spin-trapping experiments[40] (see "Supplementary Methods 3", Supplementary Fig. 10 and Supplementary Table 2), together with the mass fragmentation patterns of TPrA (Supplementary Figs. 11 and 12 and Supplementary Table 3), indicated the generation of an additional product in parallel to the radical cation, in which the amine oxidation process also led to the concomitant cleavage of a C–N bond (Fig. 3a), thereby generating an N-centered

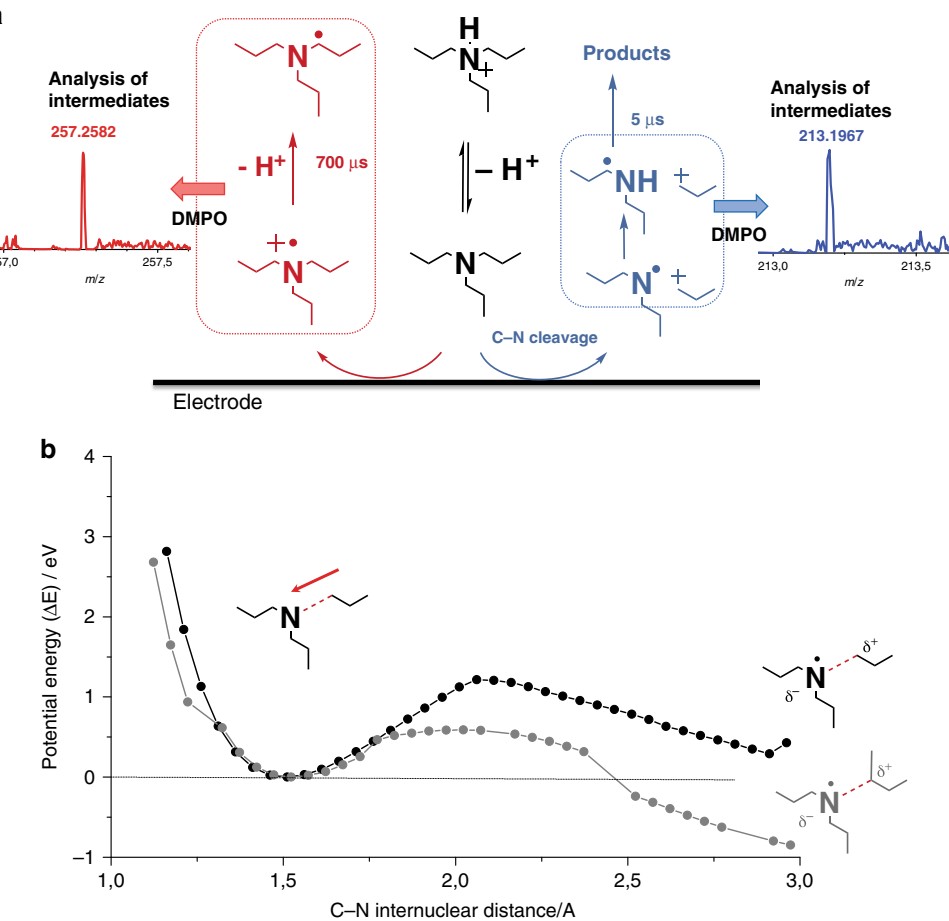

**Fig. 3 Proposed mechanism for tri-n-propylamine (TPrA) and dipropylamine (DPrA) radical generation. a** Schematic representation of the proposed parallel pathways for the tri-*n*-propylamine (TPrA) oxidation at the electrode where TPrA radical cation and radical are generated (red pathway) and where dipropylamine radical (DPrA) is generated (blue pathway). The scheme reaction is supported by spin-trapping experiments with 5,5-dimethyl-pyrroline *N*-oxide (DMPO), which stabilized the radicals and allowed identification by mass spectrometry analysis (MS) and electron paramagnetic resonance (EPR). The inset in **a** shows the MS analysis for the possible adducts DMPO-TPrA and DMPO-DPrA (see "Supplementary Methods 3" for the detailed mechanism and Supplementary Fig. 12 and Supplementary Tables 2 and 3). **b** Potential energy curve along C–N (dashed red bond in the depicted molecular structures) stretching of TPrA (black curve and dots) and DPIBA (gray curve and dots) computed with UM062X/6-31G* in $H_2O$ (described with IEFPCM method at each point of the curve, the energy of the equilibrium structure has been subtracted). The calculations are run in the presence of an external electric field (~$10^8$ V cm$^{-1}$) whose orientation is represented by the red arrow. Under the effect of the external electric field the curves acquire a typical reaction profile. The mechanism shows a prevailing ionic character with the formation of a carbocation propyl fragment (represented by the $\delta+$ charge on the molecular structures).

dipropylamine radical that could convert into a C-centered radical (see "Supplementary Methods 3")[41–44]. Stability of the amine with respect to the cleavage of the C–N bond close to the electrode was estimated via quantum chemical calculations (Fig. 3b, bottom panel). The potential energy curve (PES) of the C–N bond dissociation was computed under the effect of a strong external electric field[45], which is typical during inner sphere electron transfer (of the order of $10^8$ Vcm$^{-1}$)[46]; this reflects the potential experienced by molecules that reach the electrified interface (further computational details are provided in "Supplementary Methods 4" and Supplementary Tables 4–6). External electric fields can induce mechanistic changes in reactions[47], similar to those in an electrochemical cell[48]. Interestingly, regarding the bond dissociation, the presence of an electric field (Supplementary Figs. 13–16) in the calculation led to a noticeable difference in the bond cleavage behavior from the one in absence of an electric field. The elongation along the C–N bond acquires a typical reaction profile with a limited energy barrier to be

overcome (around 1 eV for TPrA); moreover, the mechanism shows a prevailing ionic character, suggesting the formation of a carbocation propyl fragment and a radical N fragment.

Our experiments identified two regions of reactivity, unveiling an active pathway at small distances with an efficiency that was ~10 times higher than that of the mechanism observed at large distances (Fig. 2d).

Based on these findings, we propose an additive that generates a stable carbocation, increasing the efficiency of C–N cleavage, i.e., the efficiency of the pathway at short distances. In this context, we selected the particular branched amine DPIBA[49] in combination with TPrA to maintain efficient ECL emission at large distances as well. We evaluated the effect of this additive on ECL generation by measuring ECL emission at the single-bead level in a surface generation–bead emission configuration. Figure 4 presents ECL profile emissions from 2.8-μm Ru@beads with 180 mM TPrA in a phosphate buffer (PB) solution with or without 50 mM DPIBA as an additive. The additive positively

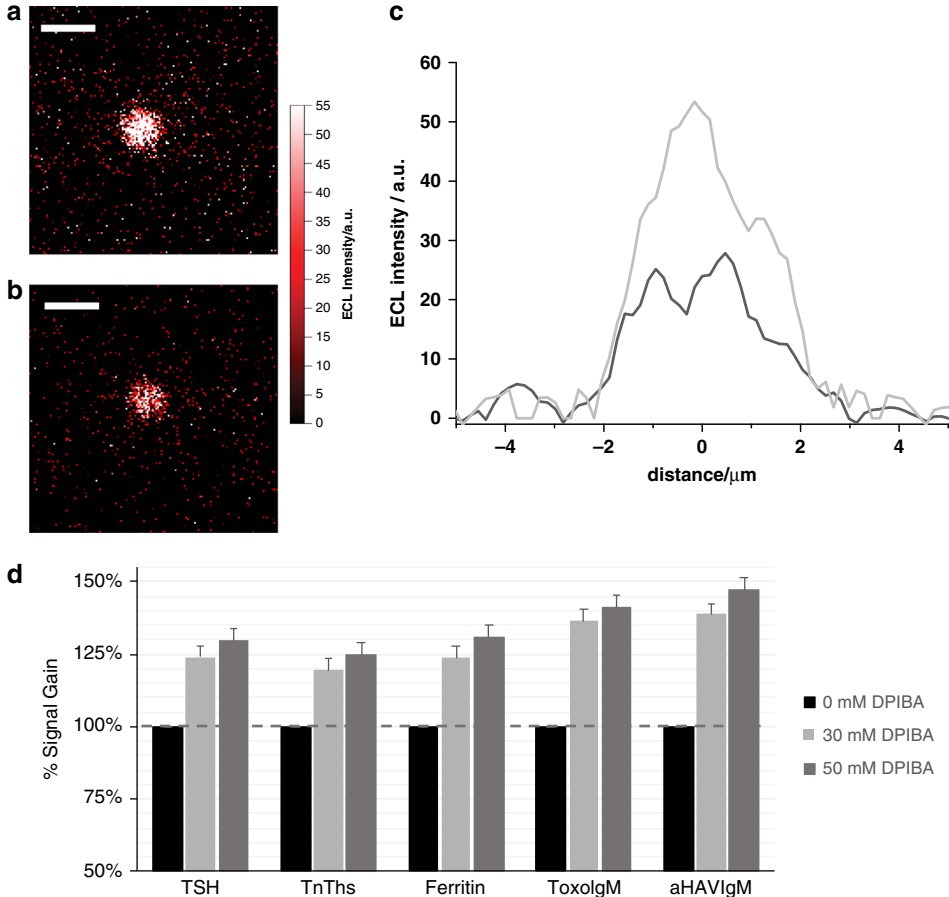

**Fig. 4 Bead-based assay and commercial immunoassay using N-dipropyl isobutyl amine (DPIBA).** Electrochemiluminescence (ECL) imaging of a 2.8-µm single bead was obtained by applying a constant potential of 1.4 V (vs. Ag/AgCl) for 4 s in 180 mM tri-*n*-propylamine (TPrA) and 0.2 M phosphate buffer (PB) with **a** 50 mM DPIBA and **b** without DPIBA. Integration time, 4 s; magnification, ×100; Scale bar, 5 µm. **c** Comparison of the bead profile lines (black, without DPIBA; and gray, with DPIBA). **d** ECL signal gain observed in the presence of 30 and 50 mM DPIBA in 180 mM TPrA, 0.2 M PB, and 0.1% polidocanol compared with a reference buffer without DPIBA, as measured on a Roche Diagnostics Cobas e 801 immunoassay analyzer using the Elecsys® assays thyroid stimulating hormone (TSH), cardiac troponin T (TnT hs), Ferritin, *Toxoplasma gondii* IgM (Toxo IgM), and hepatitis A IgM (A-HAV IgM) and a biomarker containing calibrator as sample. Error bar shows the standard deviation ($n = 10$).

affected ECL efficiency, increasing the signal by 66% with respect to TPrA alone (Supplementary Fig. 17). In addition, an even higher signal gain was observed at small distances, i.e., using 0.3-µm Ru@beads with 30 mM DPIBA as an additive (Supplementary Fig. 17). The latter result confirmed the enhancement of reactivity at small distances, a setting in which DPIBA is even more efficient, with an increase in signal of 128% compared with the standard TPrA. Control experiments using an equivalent total amine concentration in terms of TPrA showed no significant enhancement compared with the standard 180 mM TPrA (Supplementary Fig. 18), whereas DPIBA alone resulted in a lower ECL signal than TPrA (Supplementary Fig. 19).

Altogether, these experimental findings are in agreement with the computed potential energy curve of DPIBA C–N dissociation in the presence of a strong interfacial electric field (Fig. 3b, bottom panel, gray line), which exhibited a higher stability of the reaction products than that of TPrA, with an even lower energy barrier to be overcome. Therefore, the DPIBA C–N dissociative kinetic constant will be remarkably larger than the TPrA constant, rendering this mechanism much more efficient for DPIBA than for TPrA.

Finally, the signal-enhancing effect of DPIBA observed in the surface generation–bead emission experiments was assessed on a Roche Diagnostics Cobas e 801 immunoassay analyzer using a series of Elecsys® assays ("Supplementary Methods 5"). As mentioned above, commercial ECL-based immunoassays, such as Elecsys assays, are based on the combination of automated immunoassays using magnetic microbeads and ECL detection using the $[Ru(bpy)_3]^{2+}$/TPrA system (Fig. 5). ECL signals were generated under standard operating conditions on a Cobas e 801 analyzer using a biomarker containing the calibrator as sample. Rather than using the commercially available reagent containing TPrA (ProCell II M), various concentrations of DPIBA (0, 30, and 50 mM) in 180 mM TPrA, 0.2 M PB, and 0.1% polidocanol were used. The ECL signals generated were normalized against the reference lacking DPIBA. All assays showed an increase in ECL signal when DPIBA was used as an additive (Fig. 4d). Thus, the use of DPIBA as an additional coreactant is a notable example of how biomarker detection can benefit from our proposed mechanism. It is important to note that enhancement with the use of DPIBA could be even higher if smaller beads are used; this approach is currently under investigation in our laboratories.

In summary, we propose a paradigm of the ECL mechanism, with a direct impact on ECL efficiency. In fact, we were able to increase ECL emission by a maximum of 128% through (i) optimization of luminophore distribution by decreasing the bead size and (ii) addition of a branched amine to increase the efficiency of the coreactant mechanism. Overall, our results provide

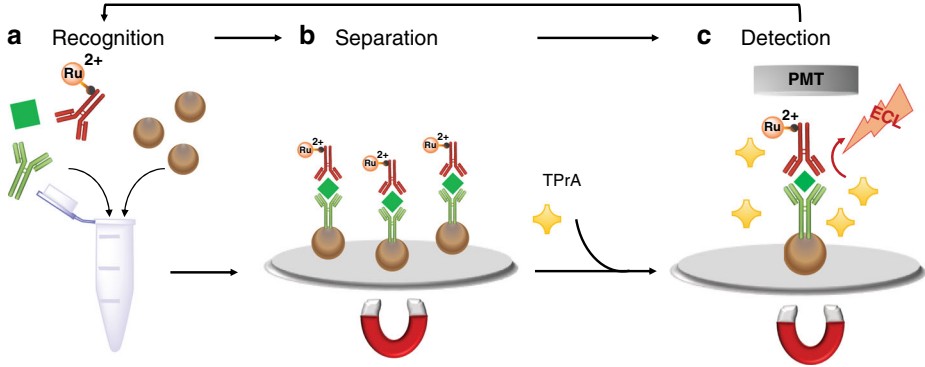

**Fig. 5 Schematic representation of the commercial electrochemiluminescence (ECL) immunoassay (sandwich assay). a** Recognition: a specific antigen (green square) is mixed together with an antigen-specific biotinylated monoclonal antibody (in green) and an antigen-specific monoclonal antibody labeled with tris(2,2′-bipyridine)ruthenium(II) (Ru$^{2+}$ in red) to form a sandwich complex which bind to streptavidin-coated microparticles (brown spheres); **b** separation: microparticles are magnetically captured onto the surface of the electrode and unspecific substances are removed; **c** detection: a voltage is applied to the electrode inducing ECL emission, which is measured by a photomultiplier tube (PMT). TPrA: tri-*n*-propylamine.

insights into the mechanisms underlying ECL generation and will pave the way for the development of highly efficient ECL coreactants for ultrasensitive biomarker analysis.

## Methods

**Chemicals**. All reagents were purchased from Sigma Aldrich, unless otherwise stated, and used without further purification. DPIBA was obtained from Roche Diagnostics (Penzberg, Germany)[49].

**Surface generation–bead emission**. Ru@beads of different sizes (0.3, 0.5, 1, and 2.8 μm) were deposited on the working electrode and collected by a magnet ("Supplementary Methods 1"). Finally, a glass cover slip was placed on top of the electrochemical cell, which was in contact with a solution of 0.2 M PB (pH 6.9) and 180 mM TPrA with or without DPIBA as an additive, as indicated. The ECL/optical imaging of surface generation–bead emission was performed using a PTFE homemade electrochemical cell comprising Pt working (0.16 cm$^2$), Pt counter, and Ag/AgCl (3 M KCl) reference electrodes. For microscopic imaging, an epifluorescence microscope from Nikon (Chiyoda, Tokyo, Japan) equipped with an ultrasensitive EMCCD camera (EM-CCD 9100–13 from Hamamatsu, Hamamatsu Japan) was used with a resolution of 512 × 512 pixels and a size of 16 × 16 μm$^2$. The microscope was enclosed in a homemade dark box to avoid interferences from external light. It was also equipped with a motorized microscope stage (Corvus, Märzhauser, Wetzlar, Germany) for sample positioning and with long-distance objectives from Nikon (20 × /0.40 DL13 mm, 100 × /0.80 DL4.5 mm). The integrated system also included a potentiostat from AUTOLAB (PGSTAT 30). Images were recorded while applying a constant potential of 1.4 V (vs. Ag/AgCl 3 M KCl) for 4 s (0.5 s for TOF calculations) with an integration time of 8 s. For TOF determination, Ru conjugated to beads (Ru@beads) was quantified by ICP-MS (X Series $^{II}$ ICP-MS from Thermo Fisher). Briefly, 500 μL of beads were dissolved in 358 μL of nitric acid (70%) and double-distilled water at a final volume of 5 mL and incubated overnight at 80 °C. After dissolution, a clear solution was obtained. The total amount of Ru, as ppb concentration, was normalized to the total surface area of each bead size to obtain the density (Ru μm$^{-2}$).

**Tip generation–surface emission**. To study the effect of distance, we used a system comprising a transparent ITO electrode functionalized with [Ru(bpy)$_3$]$^{2+}$ as the emitting surface (Ru@ITO) (see also "Supplementary Methods 2") and two different hemispherical Pt microelectrodes with diameters of 1.5 and 0.5 mm, respectively (Supplementary Figs. 6 and 8). The tip generation–surface emission experiments were performed using a PTFE homemade electrochemical cell comprising a Pt hemispherical microelectrode and Pt counter and Ag/AgCl (3 M KCl) reference electrodes in contact with a solution of 0.2 M PB (pH 6.9) and 180 mM TPrA. ITO-(from Kuramoto Seisakusho Co. Ltd., Tokyo, Japan) modified surface was positioned under an inverted microscope (Nikon) equipped with an ultrasensitive EMCCD camera (EM-CCD 9100-13 from Hamamatsu, Hamamatsu Japan) with a resolution of 512 × 512 pixels and a size of 16 × 16 μm$^2$. A 4× objective was used in all experiments. This system was connected with a CH Instruments CHI910B apparatus that accurately controlled the position in the x-, y-, and z-coordinates using stepper motor elements. All measurements were recorded by moving the electrode from a distance of 0.1–2.8 μm via the application of a potential from 0 to 1.4 V (vs. Ag/AgCl 3 M KCl) through cyclic voltammetry at a scan rate of 100 mV s$^{-1}$ and recording both ECL signals and currents. For ECL images, the system was triggered to allow the acquisition of images

in real-time during the application of potential. Images were obtained in a CCD mode sequence with an integration time of 200 ms at ×4 magnification.

**Spin-trapping and analysis of radical intermediates**. Spin-trapping experiments were conducted by adding the spin traps 5,5-dimethyl-pyrroline N-oxide (≥98%, for ESR, Sigma, 0.05 M before the electrolysis) to a solution of TPrA 180 mM, formic acid pH 6.8. The electrolysis was performed at 1.4 V (vs Ag/AgCl) for 1 h under Ar-saturated atmosphere to reduce oxygen content. Electron paramagnetic resonance (EPR) spectra were obtained using a Bruker Elexsys spectrometer operating at X-band and equipped with an ER4103 TM cavity. Mass spectrometry analysis was performed using fast flow injection (FFI) in a water/acetonitrile (50/50 + 0.1% formic acid, 50 μL/min) on a Waters Xevo G2S QTof mass spectrometer (Milford, MA, USA) equipped with an electrospray (ESI) source.

**Computational methods**. The calculations were performed using both the restricted and unrestricted density functional theory (DFT) methods, with the M062X functional[50] and the 6-31+G* basis set. The Gaussian16 suite of programs was used[51]. The calculations were performed in vacuo or including the effect of water as a solvent using the integral equation formalism model[52]. The electric field was introduced into the calculations with an orientation along the C–N bond (pointing toward the N atom) and a strength of 0.025 a.u.

**Commercial immunoassay**. A series of commercially available Elecsys® assays for the detection of specific biomarkers (TSH, troponin T, ferritin, and IgM antibodies against *Toxoplasma gondii* and hepatitis A) was used on a Cobas e 801 immunoassay analyzer to evaluate the effect of DPIBA on the assay performance ("Supplementary Methods 4" and Supplementary Table 7). The test principles of these assays are based on the formation of an immune complex between the biomarker present in the sample, a biotinylated biomarker-specific antibody, and a ruthenylated-biomarker-specific antibody (for sandwich assays)[28] or a ruthenylated antigen specific for the IgM antibody (for μ-capture assays). These immune complexes are then bound to streptavidin-coated microparticles and magnetically captured on the surface of the working electrode in the Elecsys® measuring cell. After the removal of unbound substances and the addition of a coreactant-containing buffer (typically, only TPrA is used; in this experiment, DPIBA was also used), ECL signal generation is induced via the potential application at the working electrode and detected using a photomultiplier (Fig. 5). COBAS and ELECSYS are trademarks of Roche.

## Data availability
The data that support the findings of this study are available from the corresponding authors upon request.

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

## Acknowledgements

This work is supported by the Italian Ministero dell'Istruzione, Università e Ricerca (FIRB RBAP11C58Y, PRIN-2017FJCPEX and PRIN-2017PBXPN4), University of Bologna, INSTM. S.Ra. acknowledges Fondazione AIRC per la Ricerca sul Cancro under MFAG 2016 - ID. 19044 project. We are grateful to Dr. Daniela Manzini (Centro Interdipartimentale Grandi Strumenti (CIGS) of the University of Modena and Reggio Emilia) for ICP-MS analysis, Dr. Andrea Garelli (University of Bologna) for the Gas Chromatography–Mass Spectrometry (GC-MS) analysis, and Dr. Barbara Biondi (CNR ICB, Padova) for assistance with mass spectrometry.

## Author contributions

A.Z., A.F., and S.Re. optimized the protocols, performed and analyzed the electro-chemical experiments and performed and analyzed the ECL experiments; S.C. performed the quantum mechanical calculation; A.Z., T.S., and S.Ra. performed the tip generation–surface emission experiments; N.Z. performed the commercial immu-noassays under the supervision of M. W.; H. P. synthetized and purified the additive; A.B. performed mass analysis and electrolysis product identification. T.I., K.I., F.N., M.M., G.V., and F.P. planned and supervised the research and co-wrote the paper with con-tributions from other authors. All authors discussed the results and commented the paper.

## Competing interests

The authors declare no competing interests.
