## [Peer Review File · Nature Communications]

Reviewers' comments:

Reviewer #1 (Remarks to the Author):

This is an important paper for those in the field of ECL and more widely, it should be published without delay in Nature communications. However, there are a few technical issues which should first be addressed; the grammar and phrasing is poor at the start of the paper (but improves later!) There are also a few questions to be answered/clarifications to be made.

The title should be revised because it is not sufficiently informative and not grammatically correct. I suggest something like: New insights into the mechanism of coreactant electrogenerated chemiluminescence facilitating enhanced bioanalytical performance

The phrasing and grammar in the abstract is poor, please re-write it. In particular, "multistep generation" meaning vague. "application of new chemicals" meaning vague. What is meant by "smart" in line 6? Line 7: "highlighted" Do you mean discovered? "New chemicals....bioanalysis" rephrase entirely.

P3,line7 "Research of" research into

P3 "Despite.....biomarkers" rephrase entirely

Scheme 1, Figure 1 and elsewhere: Surface generation or substrate generation. Do these phrases have different meaning?

P6: Is Ru(III) not generated directly at 1.4V?

P6. NB: Please clarify what is the relationship between integrated ECL intensity as used to calculate TOF and number of photons.

P7 & elsewhere: I would prefer TF rather than TOF (ambiguity with time of flight).

P9: In my opinion, $d = \sqrt{36Dt}$ significantly over-estimates the diffusion distance, $\sqrt{2Dt}$ will be closer to the real value. Please justify the use of $\sqrt{36Dt}$.

P12 Is the pH dependence of DPIBA likely to be different to that of TPrA, is there scope for further optimisation, please discuss briefly.

P13: NB: Possible reasons why the efficiency of excited state generation with DPIBA is higher should be discussed in terms of energetics and kinetics, even if some speculation is necessary!

Outline the proposed ECL mechanism for DPIBA

How good is the ECL efficiency of DPIBA in the absence of TPrA? C Hogan (La Trobe University)

Reviewer #2 (Remarks to the Author):

In this manuscript, the authors reported an enhanced ECL efficiency (in terms of TOF per Ru molecule) on smaller (0.3 micron) magnetic bead than that on larger (2.8 micron) bead. In a tip-approaching experiment, they further detected an exponential increase in the ECL emission when the tip was approaching to the substrate in the distance less than 1 micron. According to these results, it was hypothesized that a cleavage of C-N bond in the presence of "interfacial strong electric field" was responsible for the generation of a more efficient co-reactant, DPIBA. When the new DPIBA was added as the co-reactant in a commercial ECL assay, improved ECL intensity was detected.

While the hypothesis is interesting and of some significance, this reviewer's general feeling is that the results were very much over-interpreted. Many important flaws regarding the experiment design and control, as well as data interpretation, kept this reviewer from recommending the publication of this work. Please see the comments below.

1) In the surface generation-beads emission mode, the authors observed that the ECL emission of

0.3 μm beads was 8-fold higher than that of 2.8 μm beads, which was different from the previously reported trend. While the authors just thought about the effect of the distance away from the electrode surface, what about the effect of magnetic bead diameters? Since the properties of magnetic nanoparticles (such as optical, electrical, catalytic properties) depend strongly on their dimensions, so the authors should study the effect of different particle bead sizes (especially $< 1 \mu\text{m}$) on the ECL emission at first.

2) The authors claimed that "interfacial strong electric field" was responsible for the enhanced ECL efficiency. However, when an insulating magnetic-bead was sitting on the electrode, this reviewer is not able to see any physical (or electrochemical) basis to support a local enhancement in the electric field.

3) In addition to 2), the authors did not provide sufficient results and control experiments to support the increased cleavage of C-N bond when the magnetic bead is smaller. This should be the key evidence to support the whole manuscript. However, the authors simply provided a mass spec of two compounds (TPA and DPIBA). The present Fig. S6 is only very weakly related to the authors' statement, if not completely irrelevant. For instance, what is the yield (or efficiency) of C-N bond cleavage? What is the dependence of such yield on the size of beads?

4) Furthermore, the authors used DPIBA to increase the efficiency of C-N cleavage and the ECL signal of Ru@beads increased with the addition of DPIBA. However, the ECL of Ru@beads/DPIBA system has not been studied in this manuscript. According to the literature (U.S. Patent Application No. 16/111, 646.), the ECL signal of $[\text{Ru}(\text{bpy})_3]^{2+}/\text{DPIBA}$ system is higher than that of $[\text{Ru}(\text{bpy})_3]^{2+}/\text{TPA}$ system. Therefore, a comparison of ECL signals of Ru@beads/DPIBA system and Ru@beads/TPA system should be investigated.

5) Even if the DPIBA was able to improve the ECL intensity by 30% (Fig. 1d), it does not necessarily mean better sensitivity. Supplementary Method 4 is not sufficient at all to support the very big claim in the manuscript.

6) If the authors' statement is true, a direct deduction would be, smaller beads are beneficial for immunoassay. However, no attempts were made by the authors along this direction.

7) In page 9, "see Supplementary Figure 4" should be "see Supplementary Figure S5".

Overall, the attempt to use ECL imaging of single magnetic beads is interesting. However, the research and the present results are still too preliminary to offer sufficient supports to those big claims in the manuscript. If the authors could complete additional experiments and clarify the issues above, it may be resubmitted to this journal.

Reviewer #3 (Remarks to the Author):

In this manuscript, authors proposed a new mechanism to explain the significant enhancement of ECL efficiency of Ru@bead/TPrA system at short distances ($<1 \mu\text{m}$). On the basis of this mechanism, authors discovered that DPIBA acted as an additive to improve ECL efficiency with Elecsys assays on a Roche Diagnostics analyzer.

Authors found a significant increase of the ECL intensity ($z < 1 \mu\text{m}$) when 0.5 mm microelectrode gradually approached the ITO surface. To explain this phenomenon, authors argued that the ECL emission arises from two different mechanisms according to the tip-substrate distance. One is the classic revisited mechanism described in reference 24. Another is radical N fragment from C-N bond cleavage in electrode surface, to explain the 5 μs lifetime in Supplementary Figure 5. To demonstrate this hypothesis of C-N bond cleavage, mass analysis was performed, and PES of DPIBA and TPrA under the external electric field were computed. Moreover, the proposed new ECL mechanism involved C-N bond cleavage promotes the practical application on ECL immunoassays by the addition of DPIBA. This work is established on the basis of previous reports (ref. 24, 29, 30) and highlights the discovery of C-N bond cleavage mechanism in TPrA and other amine coreactant, which significantly enhance ECL efficiency extremely close to the electrode surface. The exploration of fundamental ECL mechanism is important in ECL field because the mechanisms can explain many complex ECL phenomena. This work is worth further consideration if the following questions are addressed properly.

1. The first question is about the ECL enhancement mechanism. In this work, authors claimed that the significant ECL enhancement extremely close to the electrode surface ($z < 1 \mu\text{m}$) arose from C-N bond cleavage mechanism. But according to the COMSOL simulation in references "Chem. Sci. 2014, 5, 2568" and "Chem. Sci. 2018, 9, 6167", Ru(bpy)₃²⁺ immobilized at particles can be oxidized through lateral charge propagation between adjacent ruthenium centers, which also resulted in the exponential enhancement of ECL efficiency ($z < 1 \mu\text{m}$). To explain whether C-N bond cleavage mechanism or lateral charge propagation between adjacent ruthenium centers led to the ECL enhancement, I think that the voltage 1.4 V in Figure 1 should be changed. If a potential less positive than that for the oxidation of Ru(bpy)₃²⁺ is used and a similar ECL enhancement is also observed, the mechanism of lateral charge propagation between adjacent ruthenium centers can be excluded.
2. The second question is about light reflection phenomenon. The smooth surface of Pt microelectrode can reflect light (shown in Supplementary Figure 3a and Supplementary Figure 4a). When the Pt microelectrode gradually approached the ITO substrate within 1 μm , the surface of Pt microelectrode can act as a mirror to reflect the ECL emission from ITO surface, and thus enhance ECL intensity collected by a 4x objective under the ITO electrode. The 4x objective cannot distinguish the ECL emission between ITO surface and Pt surface if the distance between them is less than 1 μm . Therefore, the light reflection phenomenon should be considered when comparing ECL intensity as function of tip-substrate distance. COMSOL Multiphysics can simulate ECL process and light reflection simultaneously. I recommend that authors provide COMSOL simulation to verify the experimental phenomenon if this software is available.
3. In reference 24 (Figure 7), the relationship between the ECL intensity and electrode-tip distance followed an exponential equation, but why does Figure 1d show a linear relationship (red curve) between the ECL intensity and electrode-tip distance?
4. As shown in Supplementary Figure 5, The half-life time 5 μs matches the travel time calculation for $d < 1 \mu\text{m}$. Authors ascribed 5 μs half-life time to C-N bond cleavage. The mass analysis and potential energy curve indeed demonstrated the possibility of C-N bond cleavage in thermodynamics, but how to prove that this cleavage process takes 5 μs in kinetics?
5. In the description about travel time in page 9, "see Supplementary Figure 4" should be "see Supplementary Figure 5".
6. The mass analysis demonstrated the presence of product generated by the C-N bond cleavage (Supplementary Figure 6). Authors ascribed the C-N bond cleavage to the effect of a strong external electric field (Figure 2a). However, in the revisited mechanism (reference 24), there also existed C-N bond cleavage product such as (CH₃CH₂CH₂)₂NH. How to demonstrate C-N bond cleavage product generated from external electric field rather than from the hydrolysis of P1 in eq 6?
7. Because the radical N fragment is a very strong reducing agent and can be oxidized immediately at the electrode surface under positive voltage, the concentration of radical N fragment on the electrode surface should be zero in this proposed ECL process according to reference 24. Why do authors think that radical N fragment from either TPrA or DPIBA significantly improved ECL efficiency of 0.3 μm Ru@bead at small distances even though the concentration of radical N fragment is low on the electrode surface?

8. To prove the new proposed mechanism experimentally, is it possible for authors to use other techniques to in-situ monitor the active intermediate such as radical N fragment? For example, electron spin resonance, infrared spectroscopy or Raman spectroscopy. If authors can provide compelling evidence experimentally, the proposed mechanism will be more convincing.

Overall, authors proposed a mechanism to explain the enhancement of Ru@bead near the electrode surface. The combination between this mechanism and Roche Elecsys assays promotes the revolution of clinic diagnosis potentially. This work is interesting and has potential impact on future ECL research and practical clinic diagnosis if more evidence is provided to support this mechanism.

POINT-BY-POINT RESPONSE

Manuscript NCOMMS-19-37747A

Reviewer #1

“This is an important paper for those in the field of ECL and more widely, it should be published without delay in Nature communications. However, there are a few technical issues which should first be addressed; the grammar and phrasing is poor at the start of the paper (but improves later!) There are also a few questions to be answered/clarifications to be made.”

Response: We thank the reviewer for these very positive comments.

“The title should be revised because it is not sufficiently informative and not grammatically correct. I suggest something like: New insights into the mechanism of coreactant electrogenerated chemiluminescence facilitating enhanced bioanalytical performance

The phrasing and grammar in the abstract is poor, please re-write it. In particular, “multistep generation” meaning vague. “application of new chemicals” meaning vague.

What is meant by “smart” in line 6? Line 7: “highlighted” Do you mean discovered?

“New chemicals....bioanalysis” rephrase entirely. “

Response: We thank the reviewer for the comment. Prompted by the reviewer’s comment, the manuscript was carefully reviewed by an experienced editor whose first language is English and who specializes in editing papers written by scientists whose native language is not English. We also modified the title accordingly and re-wrote the abstract entirely, as follows:

Electrochemiluminescence (ECL) is a powerful transduction technique with a leading role in the biosensing field due to its high sensitivity and low background signal. Although the intrinsic analytical strength of ECL depends critically on the overall efficiency of the mechanisms of its generation, studies aimed at enhancing the ECL signal have mostly focused on the investigation of new materials, either luminophores or coreactants, while fundamental mechanistic studies are

relatively scarce. Here, we discovered an unexpected but highly efficient mechanistic path for ECL generation close to the electrode surface (signal enhancement, 128%) using an innovative combination of ECL imaging techniques and electrochemical mapping of radical generation. Our findings, which were also supported by quantum chemical calculations, led to the identification of a family of alternative branched amine coreactants, which raised the analytical strength of ECL well beyond that of present state-of-the-art immunoassays, thus creating new opportunities for ECL applications in ultrasensitive bioanalysis.

“P3 line7 “Research of” research into “

Response: We modified the text accordingly.

“P3 Despite.....biomarkers” rephrase entirely.”

Response: We rephrased the text *“Despite the high sensitivity of ECL, the process is intrinsically surface-confined with concomitant steps involved in the generation of the analytical signal. ¹²⁻¹⁶ The optimization of the ECL mechanism is still an open problem, as this is a fundamental aspect in the operative conditions of some highly promising applications for the quantification of important biomarkers.”* as follows:

Despite its high sensitivity, however, ECL remains intrinsically a surface-confined process that incorporates concomitant steps to eventually generate the analytical signal.¹²⁻¹⁶ Optimization of such a mechanism is still underway and is of fundamental importance to several highly promising applications aimed at the quantification of crucial biomarkers.

“Scheme 1, Figure 1 and elsewhere: Surface generation or substrate generation. Do these phrases have different meaning? “

Response: We apologize for this confusion. In our manuscript “substrate generation” and “surface generation” have the same meaning; we have now unified the text by using “surface generation” exclusively.

“P6: Is Ru(III) not generated directly at 1.4V? “

Response: Regarding the objection raised by the reviewer [which was also raised by Reviewer #3 (query 1)], we agree that $\text{Ru}(\text{bpy})_3^{2+}$ is already oxidized at a potential of 1.4 V. However, in our experiment, similar to that observed in the commercial assay, $\text{Ru}(\text{bpy})_3^{2+}$ was immobilized on the particle surface, and only a relatively small fraction of dyes could be directly oxidized through lateral charge propagation between adjacent ruthenium centers. To exclude this mechanism, we repeated the “surface generation–bead emission experiment” at a lower potential (1 V) and observed the same ECL enhancement (see new Supplementary Figure S2 and the reply to Reviewer #3, query 1).

We modified the text (pages 6 and 11) and added a new Supplementary Figure S2.

Page 6: *ECL intensities were measured via imaging experiments (Figure 1a) in which a potential of 1.4 V (or 1 V, as in Supplementary Figure S2) was applied for 0.5 s...*

Page 11: *In this mechanism, the pathway that involves direct oxidation of $[\text{Ru}(\text{bpy})_3]^{2+}$ was negligible. In fact, the control experiments performed at a working potential lower than the luminophore oxidation potential (i.e., <1.2 V; Supplementary Figure S2) resulted in almost identical TOF values (vs. bead size) compared with the experiments performed at 1.4 V. These observations are also in line with the results of our previous study, in which we estimated the effect of direct $[\text{Ru}(\text{bpy})_3]^{2+}$ oxidation on ECL intensity at a very small distance (<9 nm) from the electrode surface alone.³⁸*

Supplementary Figure S2. Potential dependence. a) Cyclic voltammetry of the 0.5 mM $\text{Ru}(\text{bpy})_3^{2+}/\text{PB}$ solution showing the potential used for the surface generation–bead emission experiment. b) Surface generation–bead emission experiment performed at 1 V (red line) and at 1.4 V (blue line), respectively, before and after the potential for $\text{Ru}(\text{bpy})_3^{2+}$ oxidation.

“P6. Please clarify what is the relationship between integrated ECL intensity as used to calculate TOF and number of photons.”

Response: We described how we used ECL intensity for TOF calculations in the supporting information. To better understand how we integrated ECL signals, we added details of image acquisition in Supplementary Methods 1. We also added details of the integration procedure performed with a standard integration procedure using ImageJ.

“P7 & elsewhere: I would prefer TF rather than TOF (ambiguity with time of flight).”

Response: We thank the reviewer for the comment. We chose this abbreviation based on the terminology used in the literature (new ref. 35 has been added) and to highlight the similarity between the catalytic (e.g., enzymatic) cycle and ECL generation. We think that the similitude between catalytic cycle and ECL is an important point of our manuscript and; therefore, we would prefer to keep the abbreviation TOF. We defined TOF both in the text and in the supporting information.

“P9: In my opinion, $d = \sqrt{36Dt}$ significantly over-estimates the diffusion distance, $\sqrt{2Dt}$ will be closer to the real value. Please justify the use of $\sqrt{36Dt}$.”

Response: We used $d = \sqrt{36Dt}$ because the diffusion layer is completely contained within a distance of $6\sqrt{Dt}$ from the electrode, i.e., 99.998%. In fact, at distances much greater than the diffusion layer thickness, the electrode does not show an appreciable effect on concentrations, and the reactant molecules that are in place have no access to the electrode.

However, the electrode process is powerfully dominant when distances are much smaller.

In fact, as reported by AJ Bard, LR Faulkner, Electrochemical Methods 2nd ed., p. 164:

*“One can also see from Figure 5.2.1 that the diffusion layer, that is the zone near the electrode where concentrations differ from those of the bulk, has no definite thickness. The concentration profiles asymptotically approach their bulk values. Still, it is useful to think about the thickness in terms of $(D_0t)^{1/2}$ which has units of length and characterizes the distance that species 0 can diffuse in time t . Note that the argument of the error function in (5.2.13) is the distance from the electrode expressed in units of $2(D_0t)^{1/2}$ **When its arguments are 1, 2, and 3 (i.e., when x is 2, 4, and 6 times $(D_0t)^{1/2}$) it has values, respectively, of 0.84, 0.995, and 0.99998; thus the diffusion layer is completely contained within a distance of $6(D_0t)^{1/2}$ from the electrode.** “For most purposes, one can think of it as being somewhat thinner. People often talk of a diffusion layer thickness, because there is a need to describe the reach of the electrode process into the solution.”*

“P12 Is the pH dependence of DPIBA likely to be different to that of TPrA, is there scope for further optimization, please discuss briefly.”

Response: The pH dependence in the case of using amine as an ECL coreactant strongly depends on the pKa value of the amine because of the equilibrium between TPrA and protonated TPrA. As reported previously by Pastore and coworkers (P. Pastore et al. Electrochimica Acta 51 (2006) 5394–5401), different amines with similar pKa values, dissolved in the same acid–base system, will show similar behaviors because essentially the same reactions are present.

“P13: NB: Possible reasons why the efficiency of excited state generation with DPIBA is higher should be discussed in terms of energetics and kinetics, even if some speculation is necessary! “

Response: The efficiency of excited state generation can be discussed based on the computed potential energy surfaces shown in Figure 2 as a function of C–N bond elongation in the presence of a strong electric field. For both TPrA and DPIBA, the curve acquires a typical dissociative profile, with an activation barrier for dissociation that is significantly lower for DPIBA than for TPrA. Therefore, the DPIBA dissociative kinetic constant is expected to be remarkably larger than TPrA constant, rendering the new mechanism much more efficient for the former than for the latter. This is consistent with the experimental observations.

We modified the text accordingly (page 14):

Page 14: *Therefore, the DPIBA C–N dissociative kinetic constant will be remarkably larger than the TPrA constant, rendering this new mechanism much more efficient for DPIBA than for TPrA.*

“Outline the proposed ECL mechanism for DPIBA

How good is the ECL efficiency of DPIBA in the absence of TPrA?”

Response: We thank the reviewer for raising this important point and apologize for the omission. The ECL emission of DPIBA was tested at different concentrations (5–180 mM) and was compared with 180 mM TPrA in the surface generation–bead emission experiments. These new data confirmed the superiority of TPrA over DPIBA (50% higher) under the same conditions and at the same concentrations. Please note that the ECL intensity signals were corrected for the background, i.e., the integrated signal was measured in the absence of $[\text{Ru}(\text{bpy})_3]^{2+}$ -labeled beads.

We added graphs of ECL emission in DPIBA vs. TPrA as Supplementary Figure S15.

We also modified the text (page 14) and added a new Supplementary Figure S15:

Page 14: ...whereas *DPIBA* alone resulted in a lower ECL signal than *TPrA* (Supplementary Figure S15).

Supplementary Figure S15. ECL emission of DPIBA. Surface generation–bead emission experiment performed with a) 2.8- μm beads in 50, 100, and 180 mM DPIBA in 0.2 M phosphate buffer (pH 6.9) and b) 2.8- μm beads in 180 mM DPIBA in 0.2 M phosphate buffer (pH 6.9) and 180 mM TPrA in 0.2 M phosphate buffer (pH 6.9). Potential was applied for 4 s at 1.4 V. Magnification, 100 \times .

Reviewer #2

“In this manuscript, the authors reported an enhanced ECL efficiency (in terms of TOF per Ru molecule) on smaller (0.3 micron) magnetic bead than that on larger (2.8 micron) bead. In a tip-approaching experiment, they further detected an exponential increase in the ECL emission when the tip was approaching to the substrate in the distance less than 1 micron. According these results, it was hypothesized that a cleavage of C-N bond in the presence of “interfacial strong electric field” was responsible for the generation of a more efficient co-reactant, DPIBA. When the new DPIBA was added as the co-reactant in a commercial ECL assay, improved ECL intensity was detected.

While the hypothesis is interesting and of some significance, this reviewer’s general feeling is that, the results were very much over-interpreted. Many important flaws regarding the experiment design and control, as well as data interpretation, kept this reviewer from recommending the publication of this work. Please see the comments below.”

Response: We thank the reviewer for the general appreciation of our manuscript and for raising criticisms that prompted us to add experimental details and reformulate the discussion to improve our work and render the interpretation of our experimental data more convincing.

“1) In the surface generation-beads emission mode, the authors observed that the ECL emission of 0.3 μm beads was 8-fold higher than that of 2.8 μm beads, which was different from the previously reported forecast trend. While the authors just thought about the effect of the distance away from the electrode surface, what about the effect of magnetic beads diameters? Since the properties of magnetic nanoparticles (such as optical, electrical, catalytic properties) depend strongly on their dimensions, so the authors should study the effect of different particle bead sizes (especially $< 1 \mu\text{m}$) on the ECL emission at first.”

Response: The main result of our work was the observation that close to the electrode surface, the ECL signal increases exponentially faster than the previously reported trend. This evidence was obtained using two complementary experimental approaches called (i) “surface generation–bead emission” and (ii) “tip generation–surface emission.” In the

former, we changed the size of the beads and analyzed the ECL emission from a single bead. In this case, the beads were considered inert and were used for mapping ECL emission at different distances from the electrode. In the second experimental setup, a microelectrode tip was used for mapping ECL emission generated at different tip–surface distances from surfaces modified with luminophores.

In fact, those two experimental approaches yielded, in a completely alternative and complementary way, very similar results and highlighted, in both cases, a strong emission enhancement (8-fold) for distances of $<1 \mu\text{m}$. Implicitly, such a result would support also our assumption that beads can be treated as inert supports that do not play any active role in the generation of the observed effects.

“2) The authors claimed that “interfacial strong electric field” was responsible for the enhanced ECL efficiency. However, when an insulating magnetic-bead was sitting on the electrode, this reviewer is not able to see any physical (or electrochemical) basis to support a local enhancement in the electric field.”

Response: We thank the reviewer for raising this point, which prompted us to clarify the effect of electric field on the cleavage of the C–N bond. We agree with the reviewer regarding the inertness of the beads on the electrode. In fact, the interfacial strong electric field develops when the potential required to trigger amine oxidation is applied at the working electrode, but no specific role in the generation and distribution of such an electric field would be associated with the beads sitting on the electrode surface. We added one reference on the effect of an electric field on bond-breaking reactions at electrochemical interfaces; moreover, to explain this point better, we added a new reference (48), modified Figure 2a in the main text (page 12), and rephrased the paragraph accordingly:

Page 12: *The potential energy curve (PES) of the C–N bond dissociation was computed under the effect of a strong external electric field,⁴⁵ which is typical in the case of inner sphere electron transfer (of the order of 10^8 Vcm^{-1})⁴⁶; this reflects the potential experienced by molecules that reach the*

electrified interface (further computational details are provided in Supplementary Methods 4). External electric fields can induce mechanistic changes in reactions,⁴⁷ as in an electrochemical cell.⁴⁸

“3) In addition to 2), the authors did not provide sufficient results and control experiments to support the increased cleavage of C-N bond when the magnetic bead is smaller. This should be the key evidence to support the whole manuscript. However, the authors simply provided a mass spec of two compounds (TPA and DPIBA). The present Fig. S6 is only very weakly related to the authors’ statement, if not completely irrelevant. For instance, what is the yield (or efficiency) of C-N bond cleavage? What is the dependence of such yield on the size of beads?”

Response: In our manuscript, we used different bead sizes to study ECL emission at different distances from the electrode surface. This experimental approach allowed us to evidence a new pathway that involves the cleavage of the C–N bond and is active only at a very short distance from the electrode surface. However, beads, and in particular their sizes, are not claimed to affect the new mechanism. The computational results show that the cleavage of the C–N bond depends solely on the strength of the electric field: in the field, the potential energy curves of C–N bond elongation acquire a typical reaction profile (Figure 2), allowing the dissociation to occur.

Conversely, the presence of additives, such as DPIBA, with a faster cleavage of the C–N bond had a strong enhancement effect on the ECL efficiency because they may increase the efficiency of the mechanism with such a short distance with respect to TPrA, as shown in Supporting Figure S13. To explain this point better, we added a new reference (48), modified Figure 2a in the main text, and rephrased the paragraph at page 12.

We understand the Referee’s doubts about the significance of the reported mass spectra in this context; however, we believe that these data are relevant as the higher intensity observed for the mass fragment with m/z 100 in the case of DPIBA compared with TPrA indicates a more likely cleavage of the C–N bond in the former case. In addition, the potential energy curves of C–N bond elongation suggest a lower (relative) energy for the cleavage products generated from DPIBA compared with TPrA, which suggests faster kinetics.

However, to support further our mechanistic hypothesis, and in reply to the specific request from Reviewer 3, we performed a mass analysis after a prolonged electrolysis experiment aimed at identifying the proposed fragment under the electrolytic conditions applied in the ECL experiments. In particular, we used a radical trap to stabilize the electrogenerated intermediates and analyzed the mass spectrum of the solution (see new Supplementary Method 3). From this experiment, we observed the mass of the products obtained from the TPrA radical generation and from the dipropylamine radical (DPrA). In addition these experiments allow to estimate the yield of the reaction by analyzing the intensity of mass peaks obtained from the TPrA oxidation under the hypothesis of similar reactivity of the two species. From the Supplementary Figure S10 we estimated a yield for the formation of DPrA one order of magnitude lower than the yield of the TPrA radicals.

We modified the text (pages 12 and 14) and added a new Supplementary Figure S10 and added new Supplementary Method 3:

Page 12: *Analysis of products generated after prolonged electrolysis and spin-trapping experiments⁴⁰ (see Supplementary Methods 3 and Supplementary Figure S10), together with the mass fragmentation patterns of TPrA (Supplementary Figure S11), indicated the generation of an additional product in parallel to the radical cation, in which the amine oxidation process also led to the concomitant cleavage of a C–N bond (Figure 2a), thus generating an N-centered dipropylamine radical (DPrA) able to convert into a C-centered radical (see Supplementary Methods 3).^{41–44}*

Page 14: *Therefore the DPIBA C–N dissociative kinetic constant will be remarkably larger than for TPrA, making the new mechanism much more efficient for the former than for the latter.*

Supplementary Figure S10. Spin Trapping experiments and analysis of radical intermediates. Analysis of the adducts formed during 1h electrolysis at 1.4V vs Ag/AgCl 1 mL solution in Ar saturated solution of DMPO (=0.05 M)/TPA (0.18 M) in formiate buffer at pH=6.8. a) EPR experimental spectrum (black line) and simulated (red lines) as sum of three adducts (accordingly with Supplementary table 2): DMPO-OH (O)+RNO[•] (x) + either an adduct compatible with an amine radical N-centered (Add-N) or a C-centered (Add-C) (*). For reference, the spectrum of the solution of just the distilled DMPO exposed to air

for 1 h is shown at the bottom (blue line). b) and c) Mass spectrum of a solution showing the formation of spin adducts DMPO-DPrA(C)H⁺ (m/z=213.1961) and DMPOH-TPrA (+H)⁺ (m/z=257.2587); this label indicates the charged forms of DMPOH-TPrA(N)⁺ and DMPOH-TPrA(C)-H⁺ with an extra proton (see supplementary methods 3 for details).

“4) Furthermore, the authors used DPIBA to increase the efficiency of C-N cleavage and the ECL signal of Ru@beads increased with the addition of DPIBA. However, the ECL of Ru@beads/DPIBA system have not been studied in this manuscript. According to the literature (U.S. Patent Application No. 16/111, 646.), the ECL signal of [Ru(bpy)3]2+/DPIBA system is higher than that of [Ru(bpy)3]2+/TPA system. Therefore, a comparison of ECL signals of Ru@beads/DPIBA system and Ru@beads/TPA system should be investigated.”

Response: We thank the reviewer for raising this important point and we apologize for the omission. The patent mentioned by the reviewer (US version of our reference 49 patented by some of us) is focus on the effect of DPIBA on the ECL background rather than on the ECL emission enhancement. Despite of this, as the referee required, we tested the ECL emission of DPIBA at different concentrations (5–180 mM) and we compared it with the emission at 180 mM TPrA obtained in the surface generation–bead emission experiments. New data corroborate the superiority of TPrA vs. DPIBA (50% higher) at the same conditions and concentrations.

We added graphs of ECL emission in DPIBA vs. TPrA as Supplementary Figure S15.

We also modified the text (page 14 and page 13) and added a new Supplementary Figure S15:

Page 13: *Based on these findings, we propose a novel additive that generates a stable carbocation, increasing the efficiency of C–N cleavage, i.e., the efficiency of the pathway at short distances. In this context, we selected the particular branched amine DPIBA⁴⁹ in a combination with TPrA to maintain efficient also the ECL emission at large distances.*

Page 14: ...whereas DPIBA alone resulted in a lower ECL signal compared with TPrA (Supplementary Figure S15).

Supplementary Figure S15. ECL emission of DPIBA. Surface generation–bead emission experiment performed with a) 2.8- μm beads in 50, 100, and 180 mM DPIBA in 0.2 M phosphate buffer (pH 6.9) and b) 2.8- μm beads in 180 mM DPIBA in 0.2 M phosphate buffer (pH 6.9) and 180 mM TPrA in 0.2 M phosphate buffer (pH 6.9). Potential was applied for 4 s at 1.4 V. Magnification, 100 \times .

“5) Even if the DPIBA was able to improve the ECL intensity by 30% (Fig. 1d), it does not necessarily mean better sensitivity. Supplementary Method 4 is not sufficient at all to support the very big claim in the manuscript.”

Response: In our work, we tested the effect of DPIBA as an additive in the commercial instrumentation for five different markers using TPrA as a comparison. In all cases, we observed an increase in the signal of 20% to 40% (Figure 3d). We agree with the referee, we didn't test the sensitivity of the methods, but we rather focus on the analytical signal in terms of “signal gain” and “signal to noise ratio”. In compliance with the referee request, in this new manuscript's version, we did not claim a better sensitivity of our methods. In addition, we added reference 28 (Sojic, N. *Analytical Electrogenerated Chemiluminescence: From Fundamentals to Bioassays. Detection Science.* (Royal Society of Chemistry (RSC) Publishing, 2020)) that describe in detail the ECL based commercial immunoassay.

“6) If the authors’ statement is true, a direct deduction would be, smaller beads are beneficial for immunoassay. However, no attempts were made by the authors along this direction.”

Response: As pointed out by the referee, we would expected a strong effect of smaller beads (i.e., 0.3 μm) in the results obtained using the an immunoassay analyzer. However, the currently available commercial analyzers are optimized for large beads (2,8 μm) and the change in their size would affect many of the experimental parameters, in turn affecting, in a complementary way, the overall ECL emission. The analysis of the effect of bead size on the immunoassay analyzer cannot be performed without the engineering of all of the steps included in the analytical protocols, which are not directly connected with the ECL mechanism, such as bead homogenization, bead capture, bead distribution on the electrode surface, antibody recognition, etc. This is currently under investigation in our laboratories.

We modified the text to highlight this point (page 16):

Page 16: *It is important to note that enhancement with the use of DPIBA could be even higher if smaller beads are used; this approach is currently under investigation in our laboratories.*

“7) In page 9, “see Supplementary Figure 4” should be “see Supplementary Figure S5”.”

We modified the text accordingly.

Reviewer #3

“In this manuscript, authors proposed a new mechanism to explain the significant enhancement of ECL efficiency of Ru@bead/TPrA system at short distances ($<1 \mu\text{m}$). On the basis of this mechanism, authors discovered that DPIBA acted as an additive to improve ECL efficiency with Elecsys assays on a Roche Diagnostics analyzer.

Authors found a significant increase of the ECL intensity ($z < 1 \mu\text{m}$) when 0.5 mm microelectrode gradually approached the ITO surface. To explain this phenomenon, authors argued that the ECL emission arises from two different mechanisms according to the tip-substrate distance. One is the classic revisited mechanism described in reference 24. Another is radical N fragment from C-N bond cleavage in electrode surface, to explain the 5 μs lifetime in Supplementary Figure 5. To demonstrate this hypothesis of C-N bond cleavage, mass analysis was performed, and PES of DPIBA and TPrA under the external electric field were computed. Moreover, the proposed new ECL mechanism involved C-N bond cleavage promotes the practical application on ECL immunoassays by the addition of DPIBA. This work is established on the basis of previous reports (ref. 24, 29, 30) and highlights the discovery of C-N bond cleavage mechanism in TPrA and other amine coreactant, which significantly enhance ECL efficiency extremely close to the electrode surface. The exploration of fundamental ECL mechanism is important in ECL field because the mechanisms can explain many complex ECL phenomena. This work is worth further consideration if the following questions are addressed properly.

..Overall, authors proposed a mechanism to explain the enhancement of Ru@bead near the electrode surface. The combination between this mechanism and Roche Elecsys assays promotes the revolution of clinic diagnosis potentially. This work is interesting and has potential impact on future ECL research and practical clinic diagnosis if more evidence is provided to support this mechanism.”

We thank the reviewer for the positive comments. We provided a point-by-point reply to the concerns of the reviewer below as well as additional evidence of the mechanism.

“1) The first question is about the ECL enhancement mechanism. In this work, authors claimed that the significant ECL enhancement extremely close to the electrode surface ($z < 1 \mu\text{m}$) arose from C-N bond cleavage mechanism. But according to the COMSOL simulation in references “Chem. Sci. 2014, 5, 2568” and “Chem. Sci. 2018, 9, 6167”, $\text{Ru}(\text{bpy})_3^{2+}$ immobilized at particles can be oxidized through lateral charge propagation between adjacent ruthenium centers, which also resulted in the exponential enhancement of ECL efficiency ($z < 1 \mu\text{m}$). To explain whether C-N bond cleavage mechanism or lateral charge propagation between adjacent ruthenium centers led to the ECL enhancement, I think that the voltage 1.4 V in Figure 1 should be changed. If a potential less positive than that for the oxidation of $\text{Ru}(\text{bpy})_3^{2+}$ is used and a similar ECL enhancement is also observed, the mechanism of lateral charge propagation between adjacent ruthenium centers can be excluded.”

Response: We agree with the referee that lateral charge propagation and direct oxidation of the dye may have an effect on the final ECL intensity also in the case of the surface generation–bead emission experiment. The hypothesis that electron hopping may actually occur and play a role in the generation of ECL was put forward by us in a previous work [J. Am. Chem. Soc. 2009, 131, 2260–2267] and was demonstrated more recently by a numerical model [J. Phys. Chem. C 2015, 119, 26111–26118] in which electron hopping was treated analogously to a diffusional problem, according to the theoretical model proposed by Amatore et al. [Chem. Eur. J. 2001, 7, 2206–2226]. We found that the optimal hopping height that gave the best agreement with the experimental results was $< 9 \text{ nm}$. Based on those findings, we think that the effect of lateral charge propagation and direct oxidation of the dye on the ECL emission from microparticles is very low. To confirm this hypothesis, we decided to study, as suggested by the referee, ECL emission in the surface generation–bead emission experiment using 1.0 V, i.e., a potential lower than the $\text{Ru}(\text{bpy})_3^{2+}$ direct oxidation potential (standard potential, 1.2 V). In this case, the overall ECL intensity was slightly lower (because less coreactant radicals were electrogenerated); however, the trend of the ECL as a function of the bead sizes was the same as that detected in the experiment performed at 1.4 V. This clearly demonstrates that the charge percolation effect is negligible in the present case.

We modified the main text and added a new reference and a new Supplementary Figure S2 in the Supporting Material.

Supplementary Figure S2. Potential dependence. a) Cyclic voltammetry of the of 0.5 mM $\text{Ru}(\text{bpy})_3^{2+}/\text{PB}$ solution showing the potential used for the surface generation–bead emission experiment. b) Surface generation–bead emission experiment performed at 1 V (red line) and at 1.4 V (blue line), respectively, before and after the potential for the $\text{Ru}(\text{bpy})_3^{2+}$ oxidation.

“2) The second question is about light reflection phenomenon. The smooth surface of Pt microelectrode can reflect light (shown in Supplementary Figure 3a and Supplementary Figure 4a). When the Pt microelectrode gradually approached the ITO substrate within 1 μm , the surface of Pt microelectrode can act as a mirror to reflect the ECL emission from ITO surface, and thus enhance ECL intensity collected by a 4x objective under the ITO electrode. The 4x objective cannot distinguish the ECL emission between ITO surface and Pt surface if the distance between them is less than 1 μm . Therefore, the light reflection phenomenon should be considered when comparing ECL intensity as function of tip-substrate distance. COMSOL Multiphysics can simulate ECL process and light reflection simultaneously. I recommend that authors provide COMSOL simulation to verify the experimental phenomenon if this software is available. “

Response: We thank the reviewer for raising this point, which prompted us to investigate in greater detail the effect of the tip on the ECL emission. As the referee pointed out, a 4 \times

objective cannot distinguish between ECL emission on the ITO surface and light reflection from the Pt hemispherical electrode close to the surface. To study this effect, we prepared a thin layer of rhodamine on a glass slide and investigated the photoemission of the layer through an inverted microscope. We used an electrode with a geometry and dimensions that were comparable to the ones used in the *tip generation–surface emission* experiments. In particular, we excited the layer of rhodamine using a Texas red filter (Exc: 542–582 nm and Em: 604–644 nm). As expected, the photoemission was higher under the tip because of the reflection of the Pt electrode [see new Supplementary Figure S7 (panel a)]; this effect was not observed previously. Conversely, as shown clearly in the new Supplementary Figure S7 (panel c), the contribution of the reflection did not change as a function of the tip–substrate distance. Therefore, the reflection of the tip had an effect on the overall ECL intensity but did not affect the observed trend reported in Figure S6.

We modified the main text and added a new Supplementary Figure S7 in the supporting material:

Page 8 *“In addition, control experiments showed a non-negligible and constant contribution of the Pt electrode reflection to the light collected at different tip–surface distances (Supplementary Figure S7).”*

Supplementary Figure S7. Reflection of the electrode tip. Images of fluorescence intensity acquired for a rhodamine-functionalized glass slide under the large Pt hemispherical electrode (diameter = 1.5 mm). Distance of a) 1 μm and b) 3 μm from the tip to the glass. C) Integrated intensity as a function of the tip–glass distance. Magnification, 4×; filter, Texas Red®.

“3) In reference 24 (Figure 7), the relationship between the ECL intensity and electrode-tip distance followed an exponential equation, but why does Figure 1d showed a linear relationship (red curve) between the ECL intensity and electrode-tip distance?”

Response: We apologize for the inconvenience. The red line in Figure 1d was obtained using the equation reported by Bard and coworkers (reference 25). In this figure, the red line looks linear on the Y-scale that was chosen to provide a comparison with our results. To clarify this point, we added a new supplementary figure in which the exponential trend is clearer.

We added a new Supplementary Figure S3 in the Supporting Material:

Supplementary Figure S3. Exponential increase in ECL intensity as a function of the substrate distance data adapted from reference 25.

“4) As shown in Supplementary Figure 5, The half-life time $5 \mu\text{s}$ matches the travel time calculation for $d < 1 \mu\text{m}$. Authors ascribed $5 \mu\text{s}$ half-life time to C-N bond cleavage. The mass analysis and potential energy curve indeed demonstrated the possibility of C-N bond cleavage in thermodynamics, but how to prove that this cleavage process takes $5 \mu\text{s}$ in kinetics?”

Response: We understand that our scheme was somewhat misleading. In fact, the value of $5 \mu\text{s}$ represents the stability of the radical that was electrogenerated by the cleavage of the C–N bond. The cleavage process took place in a much shorter time, which was experimentally unmeasurable, as it was likely concerted with the electron transfer (Scheme 2a). This is also supported by the potential energy curves, which showed a very low activation barrier and fast kinetics for the process described above.

We modified Figure 2a to highlight the parallel pathway.

“5) In the description about travel time in page 9, “see Supplementary Figure 4” should be “see Supplementary Figure 5”.”

Response: We modified the text accordingly.

“6) The mass analysis demonstrated the presence of product generated by the C-N bond cleavage (Supplementary Figure 6). Authors ascribed the C-N bond cleavage to the effect of a strong external electric field (Figure 2a). However, in the revisited mechanism (reference 24), there also existed C-N bond cleavage product such as $(\text{CH}_3\text{CH}_2\text{CH}_2)_2\text{NH}$. How to demonstrate C-N bond cleavage product generated from external electric field rather than from the hydrolysis of P1 in eq 6?”

Response: The generation of the C–N bond cleavage product has to follow an alternative pathway, parallel to the hydrolysis product of P1 provided in Eq. 6; otherwise, this could be produced in the whole range of the diffusion length of the TPrA radical cation (on the typical length of 3 μm), thus not yielding a shorter decay. In addition, the mass analysis demonstrated the manner in which the breakage of the C–N bond and the $(\text{CH}_3\text{CH}_2\text{CH}_2)_2\text{NH}$ product are promoted by the isobutyl chain of DPIBA.

We clarified this point in the text (page 11) and modified Figure 2a to highlight the parallel pathway.

Page 11: *“In this case, the measured lifetime of the electrogenerated species lifetime was 5 μs , and suggested suggesting a parallel mechanism triggered by TPrA oxidation.”*

“7) Because the radical N fragment is a very strong reducing agent and can be oxidized immediately at the electrode surface under positive voltage, the concentration of radical N fragment on the electrode surface should be zero in this proposed ECL process according to reference 24. Why do authors think that radical N fragment from either TPrA or DPIBA significantly improved ECL efficiency of 0.3 μm Ru@bead at small distances even though the concentration of radical N fragment is low on the electrode surface?”

Response: We agree with the reviewer regarding the concentration of radicals at the electrode surface, which must be zero, as these molecules can be easily oxidized at the electrode surface. In fact, the analysis of very small distances (e.g., 0.1 μm) revealed a drop

in the ECL signal because of the consumption of the electrogenerated radicals close to the electrode surface. Our very efficient mechanism was active in the border region between the scavenger caused by electrode oxidation (distance $<0.2 \mu\text{m}$) and the region in which the TPrA radical is active (distance $<3 \mu\text{m}$).

We added a new Supporting Figure S4 to clarify the mechanism that is active at the various distances.

Supplementary Figure S4. Spatial map of the ECL emission. ECL intensity as a function of bead dimension (blue curve and dots) highlighting the mechanism that is active at the various distances.

“8) To prove the new proposed mechanism experimentally, is it possible for authors to use other techniques to in-situ monitor the active intermediate such as radical N fragment? For example, electron spin resonance, infrared spectroscopy or Raman spectroscopy. If authors can provide compelling evidence experimentally, the proposed mechanism will be more convincing.”

Response: To support further our mechanistic hypothesis, we performed a mass analysis after a prolonged electrolysis experiment aimed at identifying the proposed fragment

under the electrolytic conditions applied in the ECL experiments. In particular, we decided to use a radical trap to stabilize the electrogenerated intermediates and analyzed the mass spectrum of the solution. From this experiment, we observed the mass of the products obtained from the generation of the TPrA radical and from the dissociation of the C–N bond.

We modified the text (page 12), Figure 2a, added a new Supplementary Figure S10 and added new Supplementary Method 3:

Page 12: *Analysis of products generated after prolonged electrolysis and spin-trapping experiments⁴⁰ (see Supplementary Methods 3 and Supplementary Figure S10), together with the mass fragmentation patterns of TPrA (Supplementary Figure S11), indicated the generation of an additional product in parallel to the radical cation, in which the amine oxidation process also led to the concomitant cleavage of a C–N bond (Figure 2a), thus generating an N-centered dipropylamine radical (DPrA) able to convert into a C-centered radical (see Supplementary Methods 3).^{41–44}*

Supplementary Figure S10. Spin Trapping experiments and analysis of radical intermediates. Analysis of the adducts formed during 1h electrolysis at 1.4V vs Ag/AgCl 1 mL solution in Ar saturated solution of DMPO (=0.05 M)/TPA (0.18 M) in formiate buffer at pH=6.8. a) EPR experimental spectrum (black line) and simulated (red lines) as sum of three adducts (accordingly with Supplementary table 2): DMPO-OH (O)+RNO[•] (x) + either an adduct compatible with an amine radical N-centered (Add-N) or a C-centered (Add-C) (*). For reference, the spectrum of the solution of just the distilled DMPO exposed to air

for 1 h is shown at the bottom (blue line). b) and c) Mass spectrum of a solution showing the formation of spin adducts DMPO-DPrA(C)H⁺ (m/z=213.1961) and DMPOH-TPrA (+H)⁺ (m/z=257.2587); this label indicates the charged forms of DMPOH-TPrA(N)⁺ and DMPOH-TPA(C)-H⁺ with an extra proton (see supplementary methods 3 for details).

REVIEWERS' COMMENTS:

Reviewer #1 (Remarks to the Author):

I note that the issues raised by myself and the other reviewers have been well addressed by the authors and that the changes made to the manuscript have increased its technical quality significantly. In view of its importance for the field, I believe the work should now be published.

Reviewer #2 (Remarks to the Author):

This reviewer is satisfied with the revision and has no further questions.

Reviewer #3 (Remarks to the Author):

The revised manuscript of an earlier submission has responded to the questions raised. In my opinion, the manuscript is now ready for publication.